# Chromatin is an ancient innovation conserved between Archaea and Eukarya

Ron Ammar[1,2], Dax Torti[2], Kyle Tsui[1,3], Marinella Gebbia[1,2], Tanja Durbic[2], Gary D Bader[1,2], Guri Giaever[1,3], Corey Nislow[1,2]*

[1]Department of Molecular Genetics, University of Toronto, Toronto, Canada; [2]Donnelly Centre, University of Toronto, Toronto, Canada; [3]Department of Pharmaceutical Sciences, University of Toronto, Toronto, Canada

**Abstract** The eukaryotic nucleosome is the fundamental unit of chromatin, comprising a protein octamer that wraps ~147 bp of DNA and has essential roles in DNA compaction, replication and gene expression. Nucleosomes and chromatin have historically been considered to be unique to eukaryotes, yet studies of select archaea have identified homologs of histone proteins that assemble into tetrameric nucleosomes. Here we report the first archaeal genome-wide nucleosome occupancy map, as observed in the halophile *Haloferax volcanii*. Nucleosome occupancy was compared with gene expression by compiling a comprehensive transcriptome of *Hfx. volcanii*. We found that archaeal transcripts possess hallmarks of eukaryotic chromatin structure: nucleosome-depleted regions at transcriptional start sites and conserved −1 and +1 promoter nucleosomes. Our observations demonstrate that histones and chromatin architecture evolved before the divergence of Archaea and Eukarya, suggesting that the fundamental role of chromatin in the regulation of gene expression is ancient.

*For correspondence:
corey.nislow@gmail.com

**Competing interests:** The authors have declared that no competing interests exist

**Reviewing editor**: Danny Reinberg, New York University School of Medicine, United States

## Introduction

Archaeal nucleosome core particles protect ~60 bp of DNA, approximately half that of eukaryotic nucleosomes, as demonstrated by the landmark work of *Pereira et al. (1997)*. Comparing both eukaryotic and archaeal nucleosomes, the former is an octamer composed of heterodimers of histones H2A, H2B, H3 and H4 whereas the latter histones assemble from homologs of H3 and H4 proteins (*Pereira and Reeve, 1998*; *Talbert and Henikoff, 2010*). Archaeal histones can form both homodimers and heterodimers, as well as homotetramers, whereas eukaryotic histones contain hydrophobic dimerization surfaces that restrict assembly of the octamer from H2A-H2B and H3-H4 heterodimers (*Sandman and Reeve, 2006*; *Talbert and Henikoff, 2010*).

Using single-nucleotide resolution maps of archaeal nucleosome occupancy and gene expression, we demonstrate that the architecture of archaeal chromatin and the occupancy of its nucleosomes along transcription units are conserved. We constructed a nucleosome occupancy map of the halophilic archaeon *Haloferax volcanii*, a member of the phylum euryarchaeota, originally discovered in the highly saline sediment of the Dead Sea (*Mullakhanbhai and Larsen, 1975*). The genome of *Hfx. volcanii* has an average GC content of 65% and a total genome length of 4 Mb (*Hartman et al., 2010*) composed of five circular genetic elements: a 2.8 Mb main chromosome, three smaller chromosomes pHV1, pHV3 and pHV4 and the plasmid pHV2. It is highly polyploid with ~15 genome copies during exponential growth and ~10 during stationary phase (*Breuert et al., 2006*). The histone protein of *Hfx. volcanii*, hstA (HVO_0520), has a domain architecture containing two distinct histone fold domains within the same peptide that heterodimerize similar to the *Methanopyrus kandleri* histone (HMk) (*Geer et al., 2002*; *Talbert and Henikoff, 2010*; *Marchler-Bauer et al., 2011*).

**eLife digest** Single-celled microorganisms called archaea are one of the three domains of cellular life, along with bacteria and eukaryotes. Archaea are similar to bacteria in that they do not have nuclei, but genetically they have more in common with eukaryotes. Archaea are found in a wide range of habitats including the human colon, marshlands, the ocean and extreme environments such as hot springs and salt lakes.

It has been known since the 1990s that the DNA of archaea is wrapped around histones to form complexes that closely resemble the nucleosomes found in eukaryotes, albeit with four rather than eight histone subunits. Nucleosomes are the fundamental units of chromatin, the highly-ordered and compact structure that all the DNA in a cell is packed into. Now we know exactly how many nucleosomes are present in a given cell for some eukaryotes, notably yeast, and to a good approximation we know the position of each nucleosome during a variety of metabolic states and physiological conditions. We can also quantify the nucleosome occupancy, which is measure of the length of time that the nucleosomes spend in contact with the DNA: this is a critical piece of information because it determines the level of access that other proteins, including those that regulate gene expression, have to the DNA. These advances have been driven in large part by advances in technology, notably high-density microarrays for genome wide-studies of nucleosome occupancy, and massively parallel sequencing for direct nucleosome sequencing.

Ammar et al. have used these techniques to explore how the DNA of *Haloferax volcanii*, a species of archaea that thrives in the hyper-salty waters of the Dead Sea, is organized on a genome-wide basis. Despite some clear differences between the genomes of archaea and eukaryotes—for example, genomic DNA is typically circular in archaea and linear in eukaryotes—they found that the genome of *Hfx. volcanii* is organized into chromatin in a way that is remarkably similar to that seen in all eukaryotic genomes studied to date. This is surprising given that the chromatin in eukaryotes is confined to the nucleus, whereas there are no such constraints in archaea. In particular, Ammar et al. found that those regions of the DNA near the ends of genes that mark where the transcription of the DNA into RNA should begin and end contain have lower nucleosome occupancy than other regions. Moreover, the overall level of occupancy in *Hfx. volcanii* was twice that of eukaryotes, which is what one would expect given that nucleosomes in archaea contain half as many histone subunits as nucleosomes in eukaryotes. Ammar et al. also confirmed that that the degree of nucleosome occupancy is correlated with gene expression.

These two findings—the similarities between the chromatin in archaea and eukaryotes, and the correlation between nucleosome occupancy and gene expression in archaea—raise an interesting evolutionary possibility: the initial function of nucleosomes and chromatin formation might have been for the regulation of gene expression rather than the packaging of DNA. This is consistent with two decades of research that has shown that there is an extraordinary and complex relationship between the structure of chromatin and the process of gene expression. It is possible, therefore, that as the early eukaryotes evolved, nucleosomes and chromatin started to package DNA into compact structures that, among other things, helped to prevent DNA damage, and that this subsequently enabled the early eukaryotes to flourish.

## Results

We cultured *Hfx. volcanii* in rich media containing 2 M NaCl (***Mullakhanbhai and Larsen, 1975***). Genomic DNA was cross-linked and digested with micrococcal nuclease (MNase), with cell disruption accomplished by bead-beating (***Tsui et al., 2012***). Nucleosome-bound cross-linked genomic regions are protected from MNase digestion, in contrast to the linker DNA between nucleosomes. Mononucleosome-sized (50–60 bp) DNA fragments were gel purified and libraries were sequenced on an Illumina HiSeq2000 (***Figure 1A***). Sequence reads were aligned to the published *Hfx. volcanii* DS2 genome (***Hartman et al., 2010***) to generate a genome-wide nucleosome occupancy map. Controls included crosslinked DNA without MNase digestion as well as MNase treated nucleosome-free genomic DNA. The nucleosome occupancy data was significantly different than the control MNase digest of deproteinized 'naked' genomic DNA (r = 0.071), indicating that the nucleosome map is unaffected by any potential MNase sequence bias (***Chung et al., 2010***).

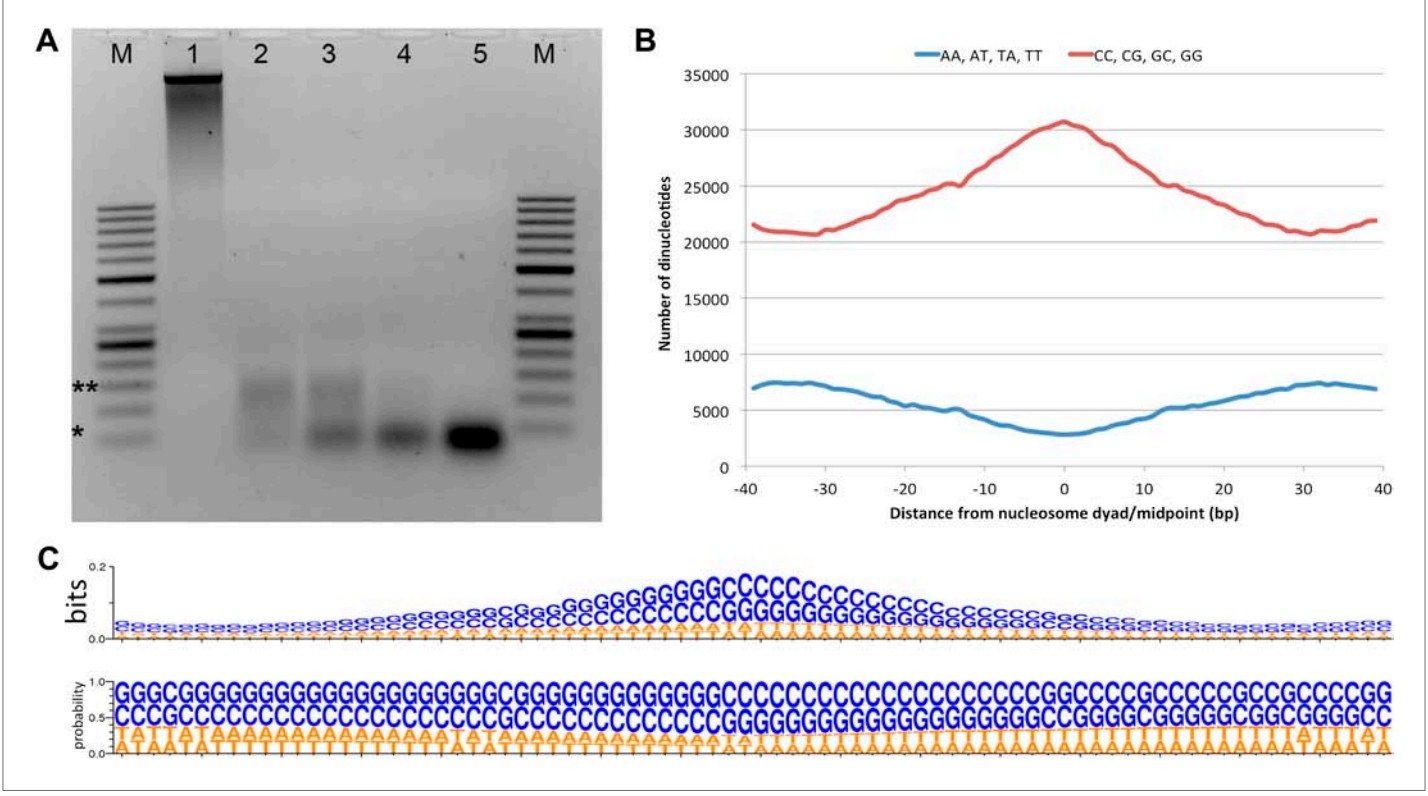

**Figure 1**. Micrococcal nuclease digestion produces nucleosomal fragments from crosslinked *Hfx. volcanii* chromatin. (**A**) Formaldehyde cross-linked chromatin was subjected to MNase digestion with increasing amounts on microccocal nuclease (from 1 to 5 units). De-crosslinked DNAs were separated on a 3% agarose gel and ~60 bp and ~120 bp mono- and di-nucleosomes were observed. Markers (M) indicate *50 bp and **150 bp. (**B**) The counts of AA, AT, TA, TT or CC, CG, GC, GG dinucleotides are reported at each position showing an enrichment of G/C nucleotides and a depletion of A/T nucleotides at the dyad relative to the end points of the protected fragment. This differs from the observation of ***Bailey et al. (2000)***, where GC, AA and TA dinucleotides were repeated at ~10-bp intervals in recombinant archaeal histone B from *Methanothermus fervidus* (rHMfB) (***Bailey et al., 2000***). (**C**) The sequence logo of a nucleosome-binding site in *Hfx. volcanii* centered at the nucleosome midpoint. There is a significant GC enrichment towards the nucleosome midpoint. This is exhibited using both bit score and probability measures.

To determine nucleosome midpoints, we smoothed the occupancy data using a symmetrical convolution sum with a Gaussian filter (***Smith, 1997***). Extrema were detected in the smoothed signal, and maxima were defined as nucleosome midpoints. In the smoothed signal, the mean peak-to-peak distance for the main chromosome was 68.5 bp in genic regions and 76.1 bp in non-genic regions. Genic regions were defined as the transcribed region plus 40 bp (the average promoter length based on ***Palmer and Daniels, 1995***) upstream of the 5' end. We observed a greater nucleosome density in *Hfx. volcanii* vs all eukaryotes likely due to the shorter length of DNA wrapped around the archaeal histone tetramer (***Pereira et al., 1997***). Based on our data, the *Hfx. volcanii* genome has 14.2 nucleosomes per kilobase compared to 5.2 nucleosomes per kilobase in *Saccharomyces cerevisiae*. The resulting map reveals a periodic pattern similar to that seen in all eukaryotes examined to date; with protected regions appearing as peaks and linker regions as troughs. Sequence analysis of the entire nucleosome map showed that nucleosome midpoints were enriched with G/C nucleotides from 61.4% GC at the edge of the protected fragment to 74.6% GC at the midpoint (dyad). We found an increase of G/C nucleotides and a decrease in A/T nucleotides at the midpoint, as described recently for human cell lines (***Figure 1B,C***) (***Valouev et al., 2011***). In contrast to previous studies in eukaryotes, we did not observe a periodicity in dinucleotide frequency relative to the nucleosome midpoint (***Satchwell et al., 1986***; ***Bailey et al., 2000***; ***Albert et al., 2007***).

We next investigated the relationship between nucleosome occupancy and gene expression. The existing genome annotation for *Haloferax* is derived almost exclusively from ORF predictions (***Hartman et al., 2010***). To augment these predictions, we used deep sequencing to create a high confidence transcriptome of the main chromosome of *Hfx. volcanii*. This map allowed us to define both 5'UTR

lengths, transcriptional start sites (TSSs) and transcriptional termination sites (TTSs). Total RNA was extracted from *Hfx. volcanii* cells, repetitive RNA was partially depleted via duplex-specific nuclease (DSN) normalization followed by RNA-seq (see 'Materials and methods') (*Zhulidov et al., 2004*). Transcript sequences were aligned, assembled and quantified using TopHat and the Genome Analysis Toolkit (*Trapnell et al., 2009*; *McKenna et al., 2010*) and transcript boundaries were further trimmed based on RNA-seq coverage information, as described previously (*Wurtzel et al., 2010*). The final set of transcripts were manually curated yielding 3059 transcriptional units in *Hfx. volcanii*, a number that is greater than observed previously in the comparable transcriptome of the sulfur-metabolizing archaeon *Sulfolobus solfataricus* (*Wurtzel et al., 2010*) but fewer than the 4073 predicted *Hfx. volcanii* genes. It is likely that in the rich media conditions used in this study, not all genes are expressed. Specifically 75% of the predicted transcripts were detectably expressed, and this fraction is consistent with observations obtained for yeast gene expression in rich media (*David et al., 2006*). Thirty-two novel transcripts (absent from the predicted sequence annotation) were identified in the RNA-seq data. Most of these 32 transcripts lack significant sequence homologs, and several were classified as transposases with paralogs in *Hfx. volcanii* (*Supplementary file 1*). Notably, the gene that was most highly expressed in the transcriptome (NTRANS_0004) was not previously annotated and contains a putative N-Acyltransferase (NAT) superfamily domain. Homology searches revealed that this transcript appears to be restricted to the genomes of other halophilic archaea (*Altschul et al., 1990*). The architecture of this domain is homologous to chain A of the well-characterized histone acetyltransferases Gcn5, Gna1, Hpa2 in *S. cerevisiae*, suggesting a possible role for this transcript in regulating transcription via histone acetylation (*Marchler-Bauer et al., 2011*). Additional acyltransferases with a similar architecture have been implicated in bacteriophage-encoded DNA modifiers as well as in cold and ethanol tolerance in yeast (*Du and Takagi, 2007*; *Kaminska and Bujnicki, 2008*). Thus, while histone post-translational modifications have not been observed in archaeal histones (*Forbes et al., 2004*), our observation suggests that some rudimentary control over chromatin accessibility may occur via the action of ancient NAT family members. Furthermore acetyltransferase and deacetylase orthologs, which appear to have enzymatic activity based on their sensitivity to the histone deacetylase (HDAC) inhibitor trichostatin A have been identified in *Hfx. volcanii* (*Altman-Price and Mevarech, 2009*). In our subsequent analysis, we focused on all genes we empirically determined to be expressed.

In eukaryotes, the TSS of the majority of expressed genes is characterized by a nucleosome-depleted region (NDR) (*Jiang and Pugh, 2009*). This NDR is flanked by the well-positioned −1 and the +1 nucleosomes. These regions direct RNA polymerase II to initiate transcription and influence the binding of promoter regulatory elements (*Jiang and Pugh, 2009*). This stereotypical pattern of nucleosome depletion at promoters and well-ordered nucleosomes in gene bodies is found in all eukaryotes, including yeast, *Drosophila*, *Arabidopsis* and humans. Using the RNA-seq-derived transcripts for *Hfx. volcanii*, we computed the degree of aggregate nucleosome occupancy for the 2343 transcripts on the main chromosome, and found that the NDR and −1 and +1 nucleosomes are conserved in *Hfx. volcanii* (*Figure 2*) suggesting that the interplay between chromatin and transcription is conserved in archaeal promoters. We generated nucleosome occupancy profiles for each transcript and clustered them hierarchically. Differential nucleosome density was observed with profiles encompassing between four to six nucleosomes in a 400-bp DNA segment spanning 200 bp on each side of the TSS (*Figure 2C*). NDRs at TTSs are also observed, and similar to those found in eukaryotes (*Lee et al., 2007*) they are less prominent than promoter NDRs in *Hfx. volcanii* (*Figure 3*).

## Discussion

Our study establishes that genome-wide nucleosome occupancy is conserved between archaea and eukaryotes (*Figure 4*). We further show that the nucleosomal protected fragments and NDRs are shorter in archaea than in eukaryotes. Our findings are particularly noteworthy because *Hfx. volcanii* likely resembles a deeply rooted ancestor that possessed eukaryotic genome architecture hallmarks such as histones, as well as bacterial hallmarks such as the Shine-Dalgarno sequence (*Sartorius-Neef and Pfeifer, 2004*). Archaeal histone tetramers likely resemble an ancestral state of chromatin, as it has been observed that functional $(H3-H4)_2$ tetramers can be formed in vitro from eukaryotic histones, and these tetramers are functional; they facilitate more rapid transcription in vitro compared to native histone octamers (*Puerta et al., 1993*). The observation that archaea contain $(H3-H4)_2$ tetramers is consistent with the proposal that formation of the canonical eukaryotic nucleosome octamer begins with $(H3-H4)_2$ tetramer assembly (*Talbert and Henikoff, 2010*).

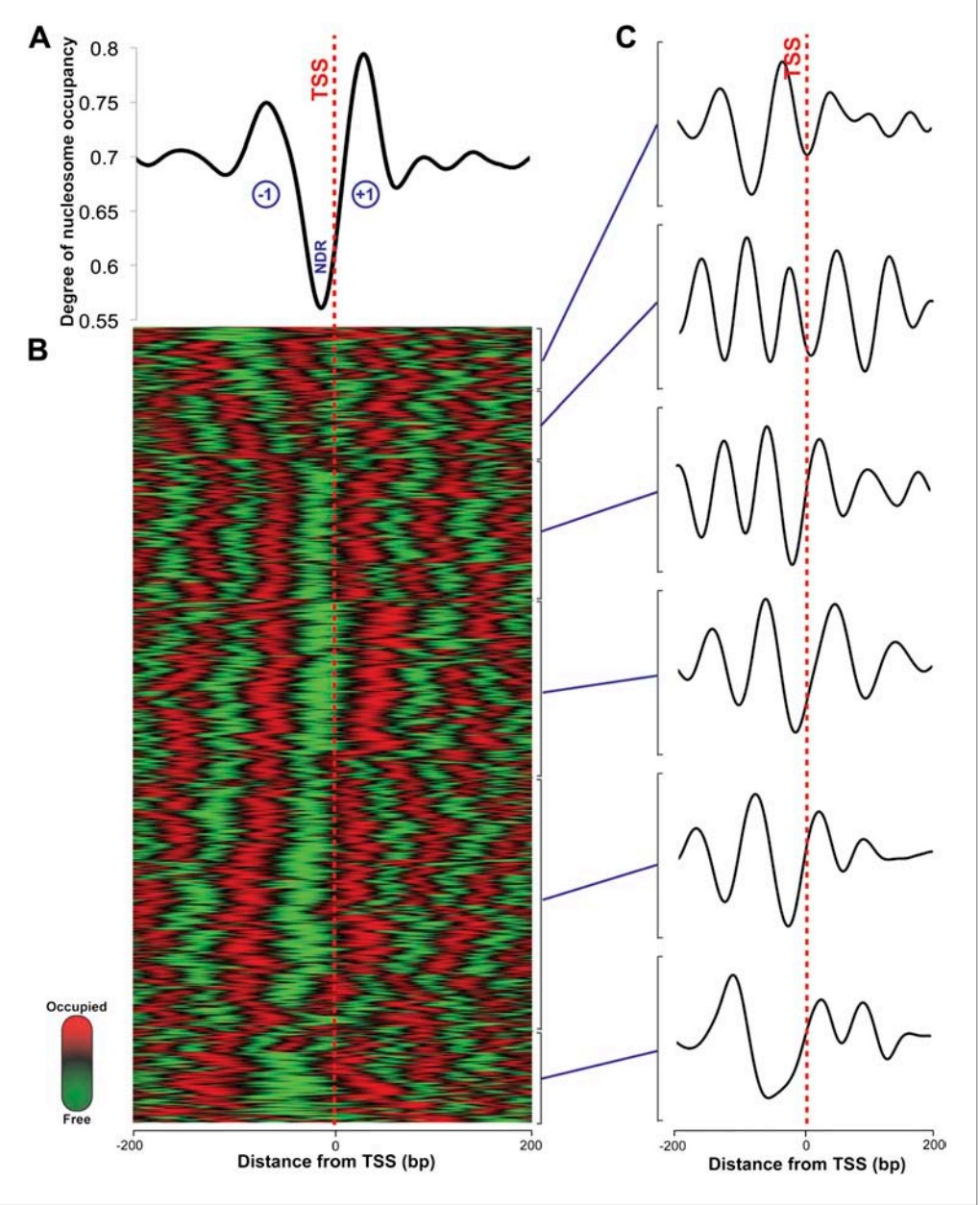

**Figure 2**. Nucleosome occupancy in *Haloferax volcanii*. (**A**) Degree of normalized nucleosome occupancy in aggregate for the main chromosome. As observed in eukaryotes, there is a prominent nucleosome-depleted region (NDR) at the transcriptional start site (TSS) preceded by a −1 nucleosome and followed by a +1 nucleosome, demonstrating that promoter genome architecture is conserved between archaea and eukaryotes. (**B**) Hierarchical clustergram for the 2343 expressed transcripts on the main *Haloferax* chromosome. Green represents nucleosome-depleted regions and red represents occupied regions. (**C**) The clustered heatmap was subdivided into the largest six subclades, and differential density of nucleosomes can be observed with occupancy profile clusters containing between four to six nucleosomes.

Our study demonstrates that both histones and chromatin architecture arose before the divergence of Archaea and Eukarya, suggesting that the fundamental role of chromatin in the regulation of gene expression is ancient. As well, owing to the small bacterial-sized archaeal genome, we suggest that archaeal chromatin is not required for genome compaction. This leads us to postulate that higher-order chromatin (*Sajan and Hawkins, 2012*) is a eukaryotic invention and that archaeal chromatin is necessary but not sufficient for genome compaction. Additionally our observations provide a rich dataset that addresses the evolution of chromatin and its fundamental role in the regulation of gene expression.

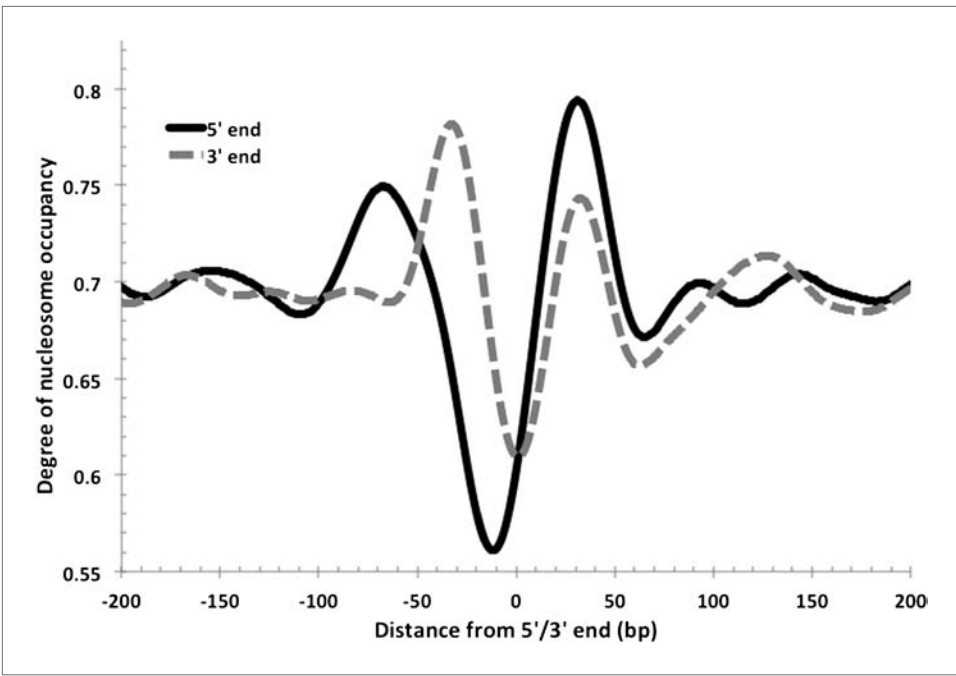

**Figure 3**. Nucleosome-depleted regions at the 5' and 3' ends of transcripts. As observed in eukaryotes, NDRs are also found at the transcriptional termination sites in *Hfx. volcanii*. Both 5' and 3' end profiles are overlaid in this figure for comparison. The 5' NDR is, on average, more depleted and longer.

## Materials and methods

### Sample preparation

*Haloferax volcanii* DS2 cells (obtained from the ATCC) were grown to mid-log phase at 42°C in ATCC 974 Halobacterium medium containing 2 M NaCl. Cells were fixed with 2% formaldehyde for 30 min then quenched with 125 mM of glycine for 5 min. An unfixed control sample was also prepared to serve as a deproteinized, 'naked' DNA control, as described previously (*Chung et al., 2010*). Cells were pelleted and snap frozen prior to MNase digestion and DNA extraction. Frozen cells were processed according to a modified protocol from *Rizzo et al. (2011)* and *Tsui et al. (2012)*. Samples were digested with increasing concentrations of MNase and a no MNase control. After digestion, fragments 50–60 bp in length were size-selected using an Agilent Bioanalyzer High Sensitivity chip (Agilent, Santa Clara, CA, part# 5067-4626) and further processed for Illumina deep sequencing. Nucleosomal and genomic libraries were pooled equally according to qPCR quantitation, and sequenced using v3 chemistry on one single-read HiSeq2000 lane (50 × 8). Samples were demultiplexed using an 8-bp index read at the end of read 1.

### Sequence read filtering and alignment

Illumina sequencers require the ligation of an adapter oligonucleotide to facilitate cluster formation on the flow cell. Because the library inserts were short (~60 bp), many sequence reads extended into the Illumina adapter sequences. The adapter subsequences were computationally trimmed to ensure maximal read mapping. Then, using a sequence quality cutoff of Phred20, reads were trimmed from both 5' and 3' ends to ensure accurate mapping. These trimmed reads from control and MNase-treated genomic DNA were aligned to the *Hfx. volcanii* DS2 genome using the Bowtie 2 gapped short read aligner (*Langmead and Salzberg, 2012*). Sequence coverage was computed using the Genome Analysis Toolkit (GATK) depth of coverage walker, which revealed the periodicity in the occupancy data (*DePristo et al., 2011*).

### Nucleosome identification

To detect nucleosome midpoint positions, sequence data were Gaussian-smoothed as described previously by *Shivaswamy et al. (2008)* and *Kaplan et al. (2009)*. This is appropriate because signals generated by processes that are random, such as sequence coverage noise, usually have a probability density function defined by a Gaussian distribution (*Smith, 1997*).

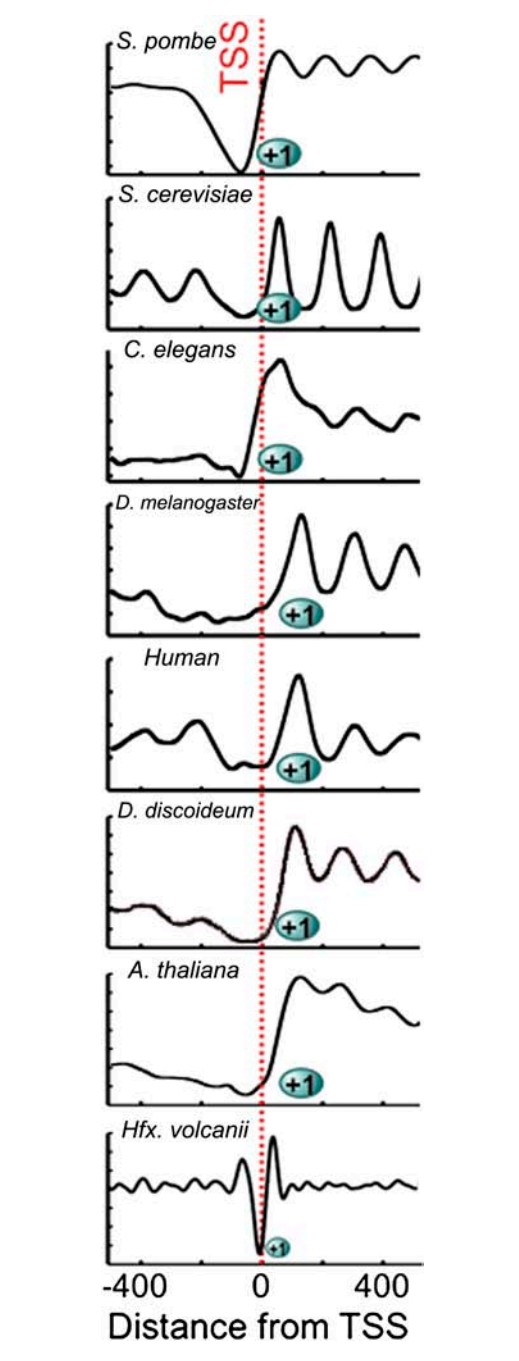

**Figure 4**. Chromatin architecture is conserved at the 5′ end of transcripts across eukaryotes and archaea. Due to the smaller size of archaeal nucleosome DNA, the occupancy has a shorter periodicity. Figure adapted with permission from **Chang et al. (2012)**.

The Gaussian filter was defined as:

$$G(x) = \frac{1}{\sqrt{2\pi}\sigma} e^{\frac{-(x-\mu)^2}{2\sigma^2}},$$

where μ is the mean of the distribution and σ is the standard deviation.

A symmetrical convolution sum was applied with the following format:

$$y[i] = \sum_{j=-\frac{M}{2}}^{\frac{M}{2}} h[j] \times x[i-j],$$

where *M* is an integer bandwidth, *y[j]* is the output, *x[j]* is the input and *h[j]* is an *M*-point function.

So, to smooth the coverage data, we applied the following convolution sum:

$$y[i] = \sum_{j=-\frac{M}{2}}^{\frac{M}{2}} G[j] \times x[i-j],$$

where $\sigma = \frac{M}{6}$. The interval length *M* is constrained to 6σ because this encompasses 99.75% of the Gaussian (**Smith, 1997**).

We also optimized nucleosome midpoint detection by convoluting a two-pass simple moving average (SMA) filter, but the Gaussian filter detected midpoints with greater resolution. Optimal interval size for the Gaussian convolution sum, as determined by Pearson's correlation coefficient with the raw data, was 27 bp. For the two-pass SMA it was 40 bp for first-pass and 15 bp for second-pass.

Nucleosome occupancy was normalized genome-wide by transforming sequence coverage data into binary-like data that existed in states of 'occupied', 'depleted' or transitioning between those two states. This final occupancy map was used to define nucleosome positions. Nucleosome occupancy profiles were clustered hierarchically by average linkage using Pearson's correlation coefficient as the similarity metric in the Cluster 3.0 software package. Clusters were visualized with Java Treeview (**Figure 2B**).

## Transcript identification and genome annotation

RNA was extracted with Trizol reagent (Invitrogen, Carlsbad, CA, 15596-026), and DNase treated (Invitrogen, Carlsbad, CA, AM1907) according to manufacturer specifications. A cDNA library was generated using 100 ng of total RNA according to Illumina TruSeq RNA Sample Prep protocol (Illumina, RS-122-2001) prior to duplex-specific nuclease (DSN) treatment. 100 ng of cDNA library was incubated in hybridization buffer (50 mM HEPES, 500 mM NaCl) for 2 min at 98°C, followed by 1 hr at 68°C. Ribosomal RNA (rRNA) was not specifically depleted (**He et al., 2010**). Instead, we used duplex-specific nuclease (DSN) normalization to remove abundant

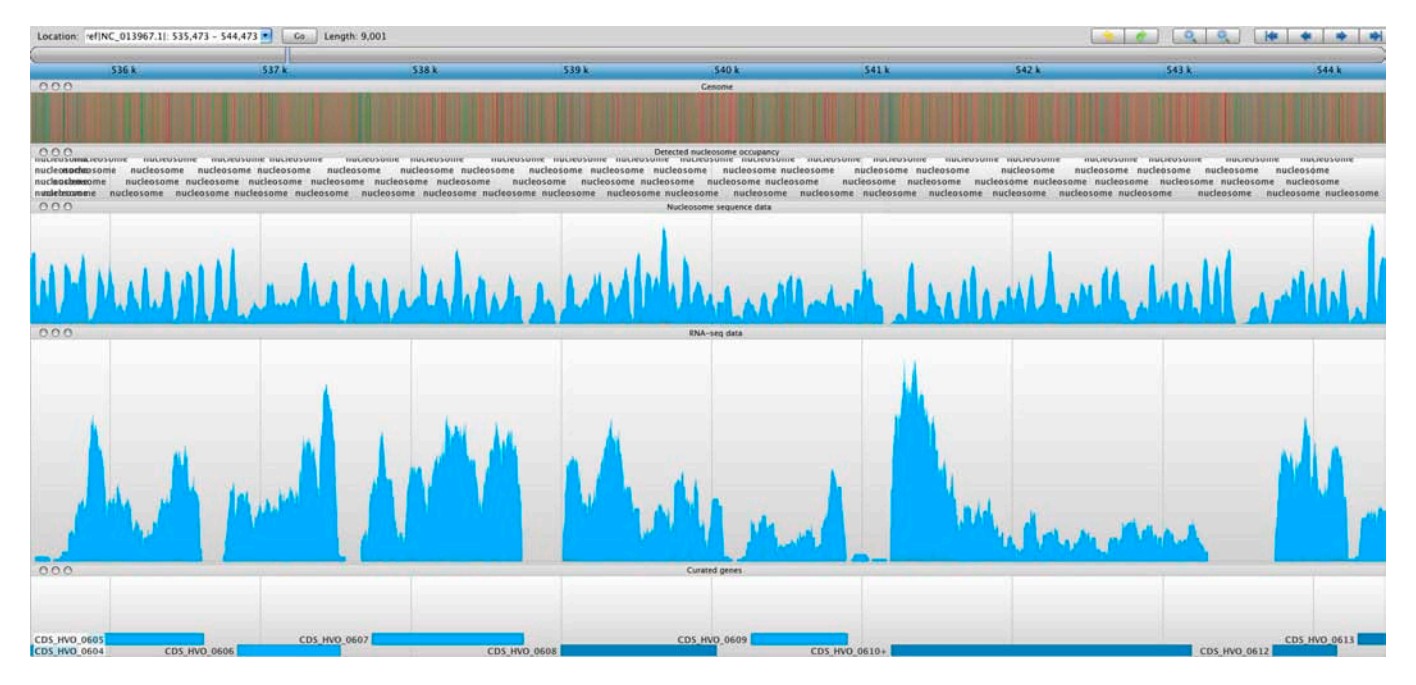

**Figure 5**. Sample screenshot of all data tracks loaded into the Savant genome browser (**Fiume et al., 2010**). The nucleosome sequence data is displayed, and the periodicity reflects protected and unprotected fragments after MNase digestion (magnitude of peak is not considered). Peaks represent nucleosome midpoints, which were detected and marked. Below are the corresponding RNA-seq and curated gene tracks. In this screenshot, one can observe seven entire ORFs in line with their NDRs and −1 and +1 nucleosomes.

RNA (rRNA, tRNA) from the total RNA sample, thereby enriching mRNA (**Zhulidov et al., 2004**). Samples were immediately treated with 4 units of DSN enzyme (Evrogen, Moscow, Russia EA001) in 1× DSN buffer and incubated for an additional 25 min at 68°C, prior to addition of stop solution, and purification with Ampure XP beads (Beckman Coulter, Brea, CA, A63881). RNA libraries were pooled equally according to qPCR quantitation, and sequenced using v3 chemistry on a paired-end single HiSeq2000 lane (100 × 8 × 100). Samples were demultiplexed using an 8-bp index read at the end of read 1. Total RNA was sequenced at extremely high coverage (2587 × mean coverage) so that rRNA sequences (~77% of all sequence reads) could be computationally excluded, as described by **Wurtzel et al. (2010)**.

After quality score trimming (described earlier), sequence reads were aligned using TopHat (**Trapnell et al., 2009**). The RNA-seq data displayed a great deal of overlap with the predicted annotations (**Hartman et al., 2010**), with 92.1% of the existing annotations being confirmed. Of the 4073 predicted annotations, 3751 were confirmed, and, of these, 744 were merged with other transcripts to form longer transcripts. A heuristic approach was applied to adjust the transcript 5′ and 3′ positions of the **Hartman et al. (2010)** predicted annotations based on the boundaries of high RNA-seq coverage regions. This was vital as TSS accuracy is of great importance for NDR identification (**Figure 5**).

Because 85% of the *Haloferax* genome is predicted to be coding (**Hartman et al., 2010**), transcript detection is complicated by transcript overlap. To overcome this, computationally identified transcripts were manually curated yielding a total of 3059 expressed transcripts in *Hfx. volcanii*. Of these, 32 transcripts are novel (**Supplementary file 1**). Of these transcripts, NTRANS_0004 was the most abundant transcript in the transcriptome, excluding the six rRNA genes. Homology data was obtained using BLASTX with a BLOSUM45 matrix against the non-redundant protein sequence database (**Altschul et al., 1990**). Conserved domains were identified using the Conserved Domain Database (**Marchler-Bauer et al., 2011**). Sequence data, nucleosome and transcriptome maps and supplemental tables have been deposited to the Short Read Archive and Dryad, as indicated in the datasets statement. Additionally this data is available at http://chemogenomics.med.utoronto.ca/supplemental/chromatin/.

## Acknowledgements
We thank H van Bakel for advice with nucleosome map and transcriptome construction.

# Additional information

### Funding

| Funder | Grant reference number | Author |
|---|---|---|
| Canadian Institutes for Health Research | MOP-86705 | Corey Nislow |
| Canadian Cancer Society | 20380 | Guri Giaever, Corey Nislow |
| Ontario Graduate Scholarship | | Ron Ammar |

The funders had no role in study design, data collection and interpretation, or the decision to submit the work for publication.

### Author contributions
RA, Conception and design, acquisition of data, analysis and interpretation of data, drafting and revising the article; DT, Acquisition of data analysis and interpretation of data; KT, Prepared samples for sequencing; MG, Prepared samples for sequencing, drafting and revising the article; Conception and design, acquisition of data analysis and interpretation of data, drafting and revising the article; TD, Prepared samples for sequencing; performed the sequencing; GB, Conception and design, analysis and interpretation of data, drafting and revising the article; GG, Interpretation of data, drafting and revising the article; CN, Directed the research, prepared biological material, conception and design, acquisition of data, analysis and interpretation of data, drafting and revising the article

# Additional files

### Supplementary files
• Supplementary file 1. A table describing the 32 novel transcripts identified in *Hfx. volcanii*.

### Major datasets

The following datasets were generated

| Author(s) | Year | Dataset title | Dataset ID and/or URL | Database, license, and accessibility information |
|---|---|---|---|---|
| Ammar R, Torti D, Tsui K, Gebbia M, Durbic T, Bader G, Giaever G, Nislow C | 2012 | Nucleosome data (MNase-seq) | SRX188663; http://www.ncbi.nlm.nih.gov/sra/SRX188663 | Publicly available at SRA (http://www.ncbi.nlm.nih.gov/sra). Reporting Standards: N/A |
| Ammar R, Torti D, Tsui K, Gebbia M, Durbic T, Bader G, Giaever G, Nislow C | 2012 | undigested DNA control | SRX188665; http://www.ncbi.nlm.nih.gov/sra/SRX188665 | Publicly available at SRA (http://www.ncbi.nlm.nih.gov/sra). Reporting Standards: N/A |
| Ammar R, Torti D, Tsui K, Gebbia M, Durbic T, Bader G, Giaever G, Nislow C | 2012 | naked DNA control (MNase-seq) | SRX185902; http://www.ncbi.nlm.nih.gov/sra/SRX185902 | Publicly available at SRA (http://www.ncbi.nlm.nih.gov/sra). Reporting Standards: N/A |

| Ammar R, Torti D, Tsui K, Gebbia M, Durbic T, Bader G, Giaever G, Nislow C | 2012 | RNA-seq | SRX188664; http://www.ncbi.nlm.nih.gov/sra/SRX188664 | Publicly available at SRA (http://www.ncbi.nlm.nih.gov/sra). Reporting Standards: N/A |
| Ammar R, Torti D, Tsui K, Gebbia M, Bader G, Giaever G, Nislow C | 2012 | Chromatin is an ancient innovation conserved between Archaea and Eukarya | http://dx.doi.org/10.5061/dryad.1qp42 | Available at Dryad Digital Repository under a CC0 Public Domain Dedication. Reporting Standards: N/A |

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
