## [Decision Letter]

Thank you for choosing to send your work entitled "Chromatin is an ancient innovation conserved between Archaea and Eukarya" for consideration at *eLife*. Your article has been evaluated by a Senior Editor and 3 reviewers, one of whom is a member of *eLife's* Board of Reviewing Editors. The following individuals responsible for the peer review of your submission want to reveal their identity: Danny Reinberg (Reviewing Editor).

The Reviewing Editor and the other reviewers discussed their comments before we reached this decision, and the Reviewing Editor has assembled the following comments based on the reviewers' reports. Our goal is to provide the essential revision requirements as a single set of instructions, so that you have a clear view of the revisions that are necessary for us to publish your work.

**General assessment and substantive concerns to be addressed during revision:**

1) The reviewers’ assessment is that the paper may contribute an important finding; however, this study lacks a critical control, which is the MNase digestion of naked DNA. MNase preferentially cleaves AT-rich DNA and the promoter regions are often AT-rich. A control MNase digestion of deproteinized genomic DNA followed by isolation of 50-60 bp length DNA for genome-wide sequencing analysis should be included. The requirement for this control MNase digestion is well documented in Chung et al. (2010) PLOS ONE 5(12): e15754 and is a pre-requisite for publication.

2) In addition, can the authors show a time course experiment where MNase digestion of *H. volcanii* genomic DNA results in multiples of 50-60 bp repetitive units? If the authors were to reconstitute chromatin in vitro using a naked plasmid and recombinant *H. volcanii* histone, how many DNA base pairs would be protected on average? Since this DNA was subject to deep sequencing, what was the actual length distribution of the DNA that was analyzed? This is important since it has not been described for this species. Of course, any other experiment demonstrating the actual size of *H. volcanii* nucleosomal DNA would be perfectly acceptable.

---

## [Author Response]

*1) The reviewers’ assessment is that the paper may contribute an important finding; however, this study lacks a critical control, which is the MNase digestion of naked DNA. MNase preferentially cleaves AT-rich DNA and the promoter regions are often AT-rich. A control MNase digestion of deproteinized genomic DNA followed by isolation of 50-60 bp length DNA for genome-wide sequencing analysis should be included. The requirement for this control MNase digestion is well documented in Chung et al. (2010) PLOS ONE 5(12): e15754 and is a pre-requisite for publication*.

In addition to the crosslinked and undigested control that we provided, we agree that the MNase digestion of deproteinized (“naked”) genomic DNA is a key control. Accordingly, we prepared *Haloferax volcanii* genomic DNA as described by Chung et al. (2010) and subjected this MNased material to high-throughput sequencing. The result shows no bias in this sample (details below).

We prepared deproteinized genomic DNA, digested it with MNase and size-selected the region of digested “naked” DNA fragments in the range of 50-60bp by gel excision, as described in Chung et al. (2010). This is exhibited in the gel images below. We show a titration of MNase- digesting “naked” DNA fragments with no nucleosomal fragment bands. The numbers above the lanes correspond to the units of MNase used, and the numbers along the DNA ladder represent the DNA length in bp.

Next, we show the size selection of “naked” DNA in the range of 50-60bp by gel excision. The numbers above the lanes correspond to the units of MNase used, and the numbers along the DNA ladder represent the DNA length in bp. The DNA from these excised bands was sequenced as a control.

The MNase-digested deproteinized DNA was aligned to the *Hfx. volcanii* genome and the profile of sequence coverage was compared to the existing nucleosome sequence coverage profile. We found that these control reads had no correlation with the nucleosome reads (Pearson’s correlation coefficient (r) = 0.071), demonstrating that the “naked” DNA control sample was not sufficient to identify nucleosome occupancy, as expected. This also indicates that the nucleosome sequence data was free from significant MNase bias. Below, we have included a screenshot displaying a representative region of genome where the nucleosome occupancy track (blue arrow) exhibits nucleosomal fragments as a trend that is significantly different from the “naked” DNA track (red arrow). These “naked” MNase control results are similar to those of Chung et al. (2010).

We have updated the manuscript to include the detail of the sample preparation and analysis, highlighting the lack of correlation between the “naked” DNA control and the nucleosomal sequence data.

*2) In addition, can the authors show a time course experiment where MNase digestion of H. volcanii genomic DNA results in multiples of 50-60 bp repetitive units? If the authors were to reconstitute chromatin in vitro using a naked plasmid and recombinant H. volcanii histone, how many DNA base pairs would be protected on average? Since this DNA was subject to deep sequencing, what was the actual length distribution of the DNA that was analyzed? This is important since it has not been described for this species. Of course, any other experiment demonstrating the actual size of H. volcanii nucleosomal DNA would be perfectly acceptable*.

We prepared *Hfx. volcanii* nucleosomal DNA as described in the manuscript, and separated the genomic material on a 3% agarose gel (new Figure 1a). This titration demonstrates the rapid digestion of high molecular weight DNA into clearly distinguishable mono- and dinucleosomes, with increased concentrations of MNase primarily digesting the material into mononucleosomal DNA. This pattern is quite similar to the digest first published by Reeve and colleagues for recombinant archaeal histone B from *Methanothermus fervidus* [Pereira SL, Grayling RA, Lurz R, & Reeve JN (1997) Archaeal nucleosomes. *Proceedings of the National Academy of Sciences of the United States of America* 94(23):12633-12637].

In addition, the length distribution of the nucleosomal DNA fragments that were subject to deep sequencing was determined using an Agilent Bioanalyzer High Sensitivity chip and is described in the Methods section.

The requested time-course experiment of MNase-digested nucleosomal fragments at multiples of 50-60bp is exhibited in new Figure 1a.

We’ve also included the Bioanalyzer electropherogram below to show that the mean fragment size of the sequenced mononucleosomal DNA fragments is 59bp for *Hfx. Volcanii*.